# Allogeneic Stem Cell Transplantation in Refractory Acute Myeloid Leukaemia

**DOI:** 10.3390/cells13090755

**Published:** 2024-04-26

**Authors:** Roberto Bono, Giuseppe Sapienza, Stefania Tringali, Cristina Rotolo, Caterina Patti, Antonino Mulè, Valeria Calafiore, Alessandra Santoro, Luca Castagna

**Affiliations:** 1BMT Unit, AOR Villa Sofia-Vincenzo Cervello, 90146 Palermo, Italy; r.bono@villasofia.it (R.B.); sapienzagius@gmail.com (G.S.); s.tringali@villasofia.it (S.T.); cristina.rotolo92@gmail.com (C.R.); 2Onco-Hematology Unit, AOR Villa Sofia-Vincenzo Cervello, 90146 Palermo, Italy; k.patti@villasofia.it (C.P.); a.mule@villasofia.it (A.M.); valeriacalaf@gmail.com (V.C.); 3Onco-Hematology and Cell Manipulation Laboratory Unit, AOR Villa Sofia-Vincenzo Cervello, 90146 Palermo, Italy; a.santoro@villasofia.it

**Keywords:** refractory acute myeloid leukaemia, allogeneic stem cell transplantation, sequential therapy, myeloablative conditioning regimen

## Abstract

Refractory acute myeloid leukaemia is very difficult to treat and represents an unmet clinical need. In recent years, new drugs and combinations of drugs have been tested in this category, with encouraging results. However, all treated patients relapsed and died from the disease. The only curative option is allogeneic transplantation through a graft from a healthy donor immune system. Using myeloablative conditioning regimens, the median overall survival regimens is 19%. Several so-called sequential induction chemotherapies followed by allogeneic transplantation conditioned by reduced intensity regimens have been developed, improving the overall survival to 25–57%. In the allogeneic transplantation field, continuous improvements in practices, particularly regarding graft versus host disease prevention, infection prevention, and treatment, have allowed us to observe improvements in survival rates. This is true mainly for patients in complete remission before transplantation and less so for refractory patients. However, full myeloablative regimens are toxic and carry a high risk of treatment-related mortality. In this review, we describe the results obtained with the different modalities used in more recent retrospective and prospective studies. Based on these findings, we speculate how allogeneic stem cell transplantation could be modified to maximise its therapeutic effect on refractory acute myeloid leukaemia.

## 1. Introduction

In acute myeloid leukaemia (AML), induction chemotherapy (IC) based on daunorubicin plus cytarabine, with or without other drugs such as FLT3 inhibitors, conjugated anti-CD3 monoclonal antibodies, or purine analogues, is associated with a high overall response rate (ORR), defined as complete remission, complete remission with incomplete haematological recovery (CRi), or a morphological leukaemia-free state (MLFS). However, as reported in Figure 1, the CR rate was different based on the European leukaemia network (ELN) risk groups, and approximatively 10–40% of patients did not achieve a complete remission/complete remission with incomplete haematological recovery after IC. An important point is to appropriately define the type of unresponsiveness, because this can have consequences on the outcome after allogeneic stem cell transplantation (allo-SCT), which represents the only curative option. The Medical Research Council (MRC) proposed an operative definition of nonresponse based on the results obtained for 8907 AML patients included in several prospective studies [1]. They classified four unresponsive groups of patients as resistant (RES), with failure to achieve complete remission (CR) after C1, partial response (PR) after first IC with fewer than 15% blasts or a greater than 50% reduction in blast percentage; refractory to first IC (REF1) with more than 15% blasts and a less than 50% proportional reduction in blast percentage; and refractory to two courses of IC (REF2). The 5-year overall survival (OS) was greater for resistant and PR patients (17% and 21%, respectively) than for patients with REF1, REF2, or REF1 + 2 (9%, 8%, and 9%, respectively). As expected, the survival of the entire cohort of unresponsive patients was lower than that of patients who achieved a CR after one course of IC (40%). Allo-SCT improved outcomes in all these categories, except for in the PR group. More recently, the ELN defined primary refractory patients as those in which no CR, or CRi was documented after 2 courses of intensive IC [2]. Usually, if able, these patients receive reinduction chemotherapy to try to achieve a CR and to proceed to allo-SCT. The CR rate with several CT combinations is lower than that after IC and ranges from 10 to 50% [3]. Targeted therapies, such as FLT3 inhibitors, are more effective than conventional chemotherapy (CT) for inducing a response in patients with FLT3+ AML. Two randomised studies comparing FLT3 inhibitors to conventional CT in refractory AML patients showed that targeted therapy induced a greater response and improved survival [4,5]. More recently, the combination of venetoclax with other CT drugs has been used in this population of patients. In a small study, the overall response rate to venetoclax plus demethylating agents or low-dose cytarabine was 21% [6]. In an Italian study, venetoclax plus a demethylating agent was administered to 190 patients, and in the RR cohort, the CR rate was 44% [7]. Finally, venetoclax was associated with conventional CT (FLAGIda) in a prospective study, and the CR rate was 67% in the phase IIB arm [8]. Based on these unexhaustive data, most refractory AML patients do not achieve a response to reinduction therapies. These refractory patients are a true challenge for haematologists. In this paper, we summarise the data from retrospective and prospective studies, and we attempt to delineate how the outcome of refractory AML can be improved. Considering the large number of retrospective studies in the literature, we considered only studies including more than 100 patients published between 2010 and 2023. Studies were searched using the following items: refractory acute myeloid leukaemia, allogeneic stem cell transplantation, and a sequential therapy, myeloablative conditioning regimen. The search was performed by LC and RB, using PubMed. LC reviewed the papers selected, excluding those published in abstract form and reviews.

## 2. Results from Retrospective Studies

In the last few decades, several retrospective studies have reported results obtained in this population. The post-transplant course of patients not in CR at transplantation is worse than that of patients in CR [26,27]. Retrospective studies in which patients received a conventional transplant modality, myeloablative (MAC) or reduced-intensity conditioning (RIC) regimens, are reported in the Table 1. The overall survival obtained ranged from 20% to 40%, based on pre- and post-transplant characteristics. The leading cause of treatment failure is almost uniformly disease progression [28,29,30,31,32,33], and the cumulative incidence of relapse (CIR) is as high as 40–50%. Furthermore, in the setting of active disease, mainly after MAC, no-relapse mortality (NRM) is also of concern, reaching 40% of patients [34,35,36]. Studies including patients with conventional allo-SCT were reported in Table 1, while those treated with a sequential approach were reported in Table 2. In the first group (Table 1), the median OS, NRM, CIR, and leukaemia-free survival (LFS) were 27%, 24%, 49%, and 32%, respectively. Most of the included patients had primary induction failure (PIF) or relapsed (either untreated or secondary), with 20% blasts at the time of transplantation. In almost all the studies, the modality of immunosuppression tapering was not reported, nor was the use of prophylactic DLIs (pDLIs), except for two studies. Craddock et al. administered a pDLI to 28 patients and a therapeutic DLI to 15 patients to treat disease relapse [37]. In another paper, DLI was given to a total of 16 patients (12%) as early as day +120. The DLI treatment was deferred by the development of aGVHD in most patients [38]. Several predictive outcome factors were identified. In Table 3, we summarise the factors influencing OS, NRM, CIR, and LFS. Several factors, such as the AML stage, blast count, and the presence of adverse cytogenetic characteristics, were recurrent in several studies. Regarding the AML stage, patients treated with fewer than three cycles of CT had better OS, which was also reflected in better LFS. PIF was associated with a better LFS and reduced risk of relapse compared with relapsed disease [39]. The presence of leukaemic cells in the bone marrow or peripheral blood was predictive of poor prognosis, and a blast count of more than 20% was associated with shorter OS and LFS. Cytogenetics has also maintained its value in patients with more advanced disease and in predicting OS, CIR, and LFS [40]. A recipient age of more than 60 years was related to survival relapse and toxicity in three studies [33,39,40]. In two studies, the intensity of the conditioning regimens was found to be significant [39,40], and, as expected, intensive regimens were more effective against leukaemia, as they improved survival rates compared to less intensive regimens. While in early studies, transplantations were allocated from matched related or unrelated donors, and, more recently, allo-SCT from mismatched related and unrelated donors was frequently performed. This approach allows us to find a donor for virtually all patients, and, because of greater HLA disparities, a greater graft-versus-leukaemia effect can be assumed, also in the setting of RR AML. Baron et al. [41] reported that there was a trend towards better OS with mis-matched unrelated donors (mMUD) than with Haplo treatment. In this registry study, GVHD prophylaxis was administered uniformly via the PTCY platform. In several studies [33,37,42], prognostic scores (Table 4) were generated, confirming the heterogeneity of these refractory patients. Indeed, in the subgroup of patients without negative factors, the OS was relatively good (38–42%), and this data can aid in selecting candidates for transplantation.

Based on the dismal results with conventional allo-SCT, a different approach using sequential conditioning followed by RIC was introduced in clinical practice [52]. Overall, the OS and LFS seemed to be slightly greater than those obtained with the conventional approach. OS ranged from 24% to 53%, and LFS ranged from 22% to 70% (Table 2). However, comparisons between different studies must be considered with caution. Two retrospective studies comparing the FLAMSA-RIC to the MAC were published in the EBMT. In the first, FLAMSA-RIC was compared to fludarabine plus treosulfan (FT) or fludarabine plus thiotepa and busulfan (TBF). The NRM, CIR, OS, and LFS were all superimposable among the three treatment groups [53]. In the second, the FLAMSA-RIC score was compared to the MAC (both TBI and BU-based) in young AML patients. Patients treated with FLAMSA were regrouped into FLAMSA and RIC groups based on CT and total body irradiation (TBI) (FLAMSA-CT and FLAMSA-TBI, respectively). No patients were reported to be infused with pDLI, but a small fraction of them received pre-emptive DLI (8% in the FLAMSA-CT cohort, 16% in the FLAMSA-TBI cohort, and 7% in the MAC cohort). The 2-year NRM was lower in the FLAMSA-CT cohort than in the other two cohorts (7% vs. 18% in the FLAMSA-TBI cohort and 16–19% in the MAC cohort), and this was confirmed by multivariate analysis. Interestingly, compared with FLAMSA-TBI and MAC, FLAMSA-CT improved OS (50% vs. 36% vs. 34%) and LFS (40% vs. 27% vs. 28–30%), with similar relapse rates. Overall, these results, within the usual limits of registry studies, suggest better survival with FLAMSA-CT in RR AML patients [47]. Prophylactic DLI is considered an important part of the sequential approach. However, the feasibility of using a pDLI is questionable in retrospective studies. Indeed, as reported in Table 2, only three studies reported or planned administration of pDLIs [45,53], limiting the role of durable CRs, while the EBMT reported the efficacy of pDLIs in reducing relapse and improving survival in patients with acute leukaemia [54]. In the original sequential approach, pDLI treatment was planned to start on day +120 if patients were in CR, without GVHD or infection, or off of immunosuppression therapy (IS). This means that only a small, selected patient population can benefit from this therapeutic intervention, introducing the concept of immortal person-time bias. To address this, Weller et al. performed a retrospective analysis using an appropriate statistical method to control for this bias. The results from this analysis confirmed that, compared with pre-emptive DLI or no intervention, the FLAMSA-RIC protocol has a more positive impact on survival [51]. However, the place of pDLI in the sequential strategy was questioned. In a prospective study from England, patients with active AML and myelodysplasia, after debulking CT, were conditioned with fludarabine and cyclophosphamide without ATG, and without planned DLI. The 2 y OS was 39% and 1 y treatment related mortality (TRM) was 33%, and 2 y cumulative incidence of relapse was 30%. In this study, the median time to relapse was 95 days, and no patients relapsed after 2 years [55].

Several predictive factors for OS, NRM, CIR, and LFS were identified in studies using a sequential approach (Table 5). Overall, older patient age (with different cut-off values) frequently had a negative impact on OS, as did poor PS and recipient CMV-positivity. A higher recipient age was also significantly associated with NRM and, as expected, with the presence of comorbidities and previous infections. Interestingly, transplantation via the unrelated donor (UD) can protect against relapse but can also increase the risk of severe toxicity, with a negative impact on OS. However, these results could be due to the time of patient accrual. Indeed, in more recent years, the outcome after transplantation has significantly improved [56], and this could also be true in the setting of RR AML.

### 2.1. Results from Prospective Studies

Few prospective studies have been conducted on RR AML patients. The main results from these experiments are reported in Table 6.

In these studies, the OS ranged from 19% to 70%, the LFS from 19% to 62%, the CIR from 20% to 54%, and the NRM from 20% to 54%. This wide range of outcomes stems from large differences in terms of inclusion criteria (age, patients, and disease characteristics), transplant technique, GVHD prophylaxis, and use of pDLI. Indeed, three studies applied a sequential approach [52,57,60], one performed allo-SCT during the aplasia period after IC [58], and two used a more conventional approach with an MAC regimen [59,61]. The pDLI treatment was planned for four studies [52,57,59,60] after treatment with different modalities to reduce the amounts of immunosuppressive drugs needed. Finally, GVHD prophylaxis was variable, even if three studies were based on cyclosporin A (CyA) plus MMF [52,57,60], and in vivo T-cell depletion was used in three trials [52,57,60], independent of the donor type. In two trials [59,61], haploidentical donors were included; in all other trials, matched donors were related or unrelated, and cord blood was matched in one [61]. Nonetheless, these differences deserve some considerations. First, allo-SCT is an effective treatment for leukemic blasts that are not sensitive to conventional chemotherapy. The sequential approach, based on debulking CT, RIC regimen, and pDLI, seems to be feasible and more effective in this situation. It is difficult to determine which part of this strategy is essential. Considering that RIC regimens exhibit low activity in refractory patients and that durable CR is less than 10% after debulking CT alone, it could be speculated that the immunological reaction exercised by pDLI is the main factor involved. However, in the three trials containing pDLI in the protocol, only a small fraction of patients received it, minimizing the weight of the beneficial results reported. On the other hand, OS was exceptionally high in patients receiving at least one pDLI. In an Indian study [59], the role of pDLI was more clear and well supported. Indeed, in one cohort of patients, the MACs and 2 y OS and LFS were 35% and 25%, respectively, but when they used MACs plus very early pDLIs (from day +21), the 2 y OS and LFS increased to 70% and 62%, respectively. However, in this study, only haploidentical donors were included, and post-transplant immunosuppressive drugs were tapered on day +14 for MMF and on day +60 for CyA. These extraordinary results need to be confirmed in more patients and in multicentre settings. Several factors predictive of outcome were identified in these studies (Table 7).

Predictive factors were different due to the heterogeneous protocols used. In the German studies [52,57], high number CD34 positive (≥9.6 × 10^6^/kg) was associated with better OS, LFS, and NRM. Furthermore, NRM was lower when the donor was matched related. Finally, less pre-treated patients (<2 induction chemotherapy) showed a better survival. The high intensity of the conditioning regimen plus the systematic infusion of donor lymphocytes can improve the survival, as well as the presence of chronic GVHD [55,58,59]. In the Italian study [61], a high comorbidities index (hematopoietic cell transplantation-comorbidity index) was associated with high NRM and low OS, while low-risk cytogenetic was associated with better OS.

### 2.2. How Can Allo-SCT Be Modified to Improve Outcomes?

Based on previous retrospective and prospective studies, it is obvious that there is room for improving the outcome of refractory AML patients treated with allo-SCT. It is clear that conventional allo-SCT performed in this setting has a small chance of being successful. To move forward from this, each segment of allo-SCT (conditioning regimen, GVHD prophylaxis, use of pDLI or pre-emptive DLI) should be re-evaluated and modified. Furthermore, the strategy of transplantation should be better scheduled (MAC vs. early sequential RIC vs. late sequential). The challenge, of course, is that every part modified in the process can have consequences for the others.

Starting from which strategy to use in a refractory AML setting, several retrospective studies and two prospective studies have shown that the relapse incidence and transplant-related mortality are high and that the OS is approximately 20% when MAC conditioning regimens are used (Table 1 and Table 6). In retrospective studies, conventional allo-SCT was basically structured as it was in patients with well-controlled disease and consisted of myeloablative conditioning regimens, regular GVHD prophylaxis, and immunosuppressive drug reduction. In one of the largest registry-based studies [42] reporting on patients transplanted from 1995 to 2004, the 3-year OS for AML patients was 19%. Several predictive factors (first CR duration ≤ 6 months, circulating blasts, donor other than HLA-identical sibling, Karnofsky or Lansky score ≤ 90, and poor-risk cytogenetics) were retrieved, identifying a subgroup of patients with a remarkable OS of 40% (Table 1). In this study, the conditioning regimens were myeloablative busulfan or TBI-based; 65% of the patients received bone marrow as a stem cell source, and GVHD prophylaxis consisted of cyclosporine A or tacrolimus plus methotrexate. In another registry-based study from Japan [38], OS was as low as that in other similar studies, with several predictive factors that can affect the outcome (Table 1). Similar results were reported by an Italian group (GITMO) in a more recent prospective study including only alternative donors conditioned with MAC regimens and with regular GVHD prophylaxis [61]. On the other hand, sequential strategies seem to improve the outcome in some but not in all studies (Table 2 and Table 6). The original protocol published by Schmid et al. has been modified in several ways, but the results were not significantly improved. Furthermore, the two approaches were compared in retrospective studies, and the clinical outcomes were not significantly different, only in one study [47] but not in the others [53,62].

The intensity of the conditioning regimen plays a role in the outcomes of AML patients [63], and, as a consequence, any improvement in terms of activity against leukaemic cells or tolerability could have a positive impact on the outcome. Thus, improving the treatment efficacy through the addition of synergistic agents may be more appealing. An alternative to this approach could be appropriate for optimising the use of regular drugs (i.e., busulfan) or other drugs. Based on in vitro studies at MD Andersson, several drugs were added to the BUFLU backbone. In a randomised trial, Alatrash et al. included 70 patients treated with different combinations of BUFLU + clofarabine. Eighty-one percent of the patients had active disease at transplantation. The OS and PFS improved in patients treated with the highest dose of clofarabine (30 mg/m^2^/day), and these improvements were not influenced by disease status. The tolerability was good, the NRM was 13%, and the CIR was 41% [64]. In another trial, the BUCY protocol was strengthened by the addition of a hypomethylating agent (decitabine 5 days before BUCY). In this study, the results obtained with decitabine added to BUCY were encouraging and better than in a historical cohort treated without decitabine [65]. In a retrospective study of the EBMT cohort, two of the most common MAC regimens were compared in an AML setting. The regimens were TBI or IV busulfan-based. Nevertheless, owing to the well-known limits of registry studies in terms of selection bias, the outcomes were similar, and the CT MAC was considered a valid alternative to the TBI MAC. Of course, the major message of this paper was addressed to those centres that did not have easy access to TBI [43]. Finally, the activity of a newer alkylating agent, such as treosulfan, was analysed and compared to the classical busulfan plus fludarabine association. Treosulfan and busulfan were administered at a myeloablative dose (42 g/m^2^ and 12.8 mg/kg, respectively). Overall, the safety profile of a treosulfan-based conditioning regimen was better, leading to an improved leukaemia-free survival and overall survival [66].

In addition to the intensity of the conditioning regimen, the proper exposure to drugs can play a central role in optimising antileukaemic activities. MD Andersson researchers were pioneers in this field. Several studies suggested that pharmacokinetic-based busulfan (BUPK) administration is more effective than the fixed-dose approach [67]. However, few centres, at least in Europe, were able to perform daily BUPK, even if the data from Houston were confirmed [68]. Another way to improve conditioning regimens without increasing toxicity is to introduce newer drugs, such as treosulfan. According to a randomised study, a reduced dose of treosulfan was more efficacious than a reduced dose of busulfan in aged patients with early-stage AML [69]. Treosulfan was also compared to TBF and FLAMSA-based in the setting of RR AML in a retrospective EBMT study [53]. In that study, treosulfan plus fludarabine (FT) had activity similar to that of TBF alone or in combination with the other sequential strategies (FLAMSA-based). However, there was a trend towards decreased early NRM and better survival in patients with FT and the sequential approach than in patients with TBF. Of course, it could be speculated that FT in a sequential scheme should improve the tolerability and activity in RR AML patients.

The use of pDLI seems to be pivotal for enhancing disease control [51,70], and it is an integral step of a sequential approach. However, in the studies where pDLI was planned, the feasibility of pDLI was low, because only one-quarter of the eligible patients received DLI [52,57,60]. There were several reasons, but these were mainly linked to the inclusion criteria, such as persistent CR, no GVHD, no IS drugs, and no infections or other complications. All these factors reduce the eligibility of patients for pDLIs and delay their infusions, exposing patients to leukaemia relapse. One way to improve the application of pDLI could be to start earlier after transplantation in parallel with IS reduction. As reported by Jaiswal, this schedule was also feasible in the context of transplantation via a haplo donor [59]. For example, cyclosporine A (CyA) could be tapered from day +60 to plan for the first pDLI at the same time to continue with CyA withdrawn on day +90, coupled with the second pDLI and third pDLI after 4–6 weeks. This schedule of pDLI for immunosuppression could improve the complexity of the relationship between time, incidence of GVHD post-DLI, timing of relapse, and incidence of GVHD after infusion [71]. Another means could be to use a modified DLI. Several Chinese groups used G-CSF-mobilised DLIs for immunosuppression, and, in a prospective trial, they showed that G-mobilised pDLIs were effective and well tolerated [72]. However, even in this study, patients were eligible if they achieved CR without GVHD. The consensus recommendations for DLI after haploidentical transplantation were published in 2020 [73]. A more sophisticated DLI could also be used to improve the efficacy/toxicity ratio. The negative selection of CD45RA+ CD3 cells was demonstrated to eliminate naïve CD3 cells responsible for GVHD [74]. In a small, proof-of-principle, prospective study, early DLI CD45RA+ depletion was well tolerated without an increase in GVHD frequency [75]. However, the anti-leukaemia activity of these compounds has not been fully analysed and, thus, must be evaluated in prospective studies. Ciurea et al. showed that prophylactic NK-DLI in high-risk AML patients was feasible and reduced the risk of relapse after haploidentical transplantation [76].

The reduction in the incidence of aGVHD via a sequential approach could be another way to improve the eligibility for prophylactic immunomodulation. Indeed, the occurrence of GVHD was one reason for not administering pDLI. Outside the haploidentical context, post-transplantation cyclophosphamide (PTCY) has been shown to be effective at reducing the incidence of GVHD in patients transplanted with matched-related or unrelated and mismatched unrelated donors [77,78,79]. Thus, PTCY plus other IS drugs can be integrated as part of an adapted approach to transplantation for RR AML patients, with the aim of reducing GVHD and allowing the timely use of the pDLI. Recently, the anti-CTLA4 inhibitor abatacept was tested in patients receiving transplants from MUDs or mMUDs, and it was found that abatacept was effective at reducing the incidence of grade 3–4 aGVHD, but the incidence of 2–4 aGVHD was still too high compared to that in historical ATG-free controls [80]. More interesting seems to be the introduction of 4-dose abatacept in the PTCY platform instead of MMF. This revised PTCY platform, after infusion of haploidentical peripheral stem cells, was promising for reducing the incidence of 2–4 and 3–4 aGVHD (17% and 4%, respectively) [81].

Finally, for the global prevention of relapse in ultrahigh-risk patients, any immunomodulatory manipulation can be associated with the use of target drugs such as azacitidine [82], anti-FLT3 drugs [83,84], and anti-IDH [85].

## 3. Conclusions

From this concise review, it is evident that the treatment of refractory AML is still an unmet clinical need and that, until recently, only allogenic stem cell transplantation has offered a chance to cure this disease. However, transplantation should be adapted to patients who are at a high risk of relapse. Each segment of the transplant process should be modified to maximise the antileukaemic effect without jeopardising tolerability.

## Figures and Tables

**Figure 1 cells-13-00755-f001:**
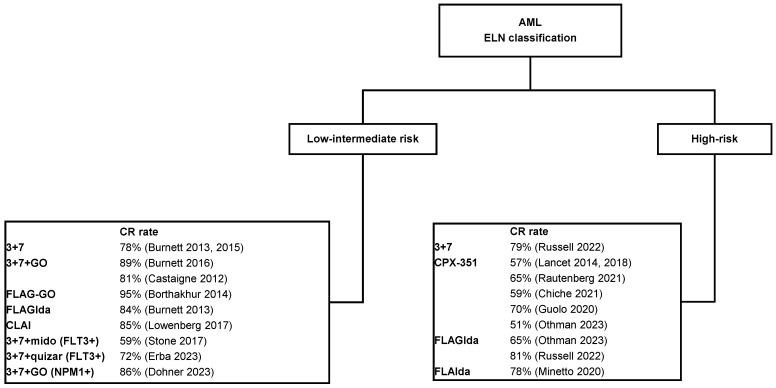
Complete remission rate in low-intermediate and high-risk AML using different induction chemotherapy. GO = gemtuzumab ozogamycin; FLAGIda: fludarabine, cytarabine, GCSF, idarubicine; CLAI: clofarabine, cytarabine, idarubicine; mido: midostaurin; quizar: quizartinib [9,10,11,12,13,14,15,16,17,18,19,20,21,22,23,24,25].

**Table 1 cells-13-00755-t001:** Studies in refractory patients treated with standard conditioning.

	N	Age	Inclusion Criteria	% Blasts (Median)	Donor Type	Conditioning	GvHD Prophylaxis	IS Tapering	pDLI	CIR	LFS	OS	NRM
Duval 2010 [42]	1673	38	PIF, untreated, refractory relapse	21%	MRD/MUD	MAC	FK/CyA +/− MTX T-cell depletion (13%)	NR	No	NR	NR	19%	38%
Craddock 2011 [36]	168	40	PIF	38%	MUD	MAC/RIC		NR	No	NR	20%@5y	22%@5y	NR
Hemmati 2014 [37]	131	52	PIF and relapse	22%	MRD/MUD	MAC/RIC FLAMSA-RIC (21%)	CyA + MTX/MMF	MRD +30 MUD +60	Yes	48%@5y	25%@5y	/	26%@3y
Liu 2015 [40]	133	40, 30 21	PIF and relapse	26%	MRD/MUD Haplo	MAC	CyA + MTX CyA + MTX + MMF GIAC protocol	NR	No	NR	36%@3y	40%@3y	19%@3y
Nagler 2015 [43]	852	43 39	PIF and relapse	20% 16%	MRD/MUD	BUCY TBICY	CyA + MTX ATG	NR	No	53%@2y 54%@2y	25%@2y 28%@2y	31%@2y 33%@2y	21%@2y 17%@2y
Todisco 2017 [33]	227	49	PIF	>25%	MRD/MUD Haplo CB	MAC 69% RIC 31%	T-cell depletion 50%	NR	No	61%@3y	23%@y	14%@3y	27%@3y
Nagler 2022 [39]	3430	55	PIF and relapse	NR	MRD/MUDHaplo	MAC 54% RIC 46% FLAMSA-RIC 13%	CyA + MTX/MMF ATG 78% PTCY 4%	NR	no	48%@2y	28%@2y	36%@2y	24%@2y
Baron 2022 [41]	219	56	PIF and relapse	NR	mMUD/Haplo	MAC/RIC	PTCY-based	NR	no	40%@2y 50%@2y	42%@2y 26%@2y	46%@2y 28%@2y	18%@2y 24%@2y
Yanada 2023 [38]	6927	53	PIF and relapse	NR	MRD/MUD CB	MAC 67% RIC 33%	FK/CyA-based	NR	no	53%@5y	NR	23%@5y	27%@5y

PIF = primary induction failure; MRD = matched related donor; MUD = matched unrelated donor; mMUD = mis-matched unrelated donor; MAC = myeloablative conditioning regimens; RIC = reduced intensity conditioning regimens; FK = tacrolimus; CyA = cyclosporine A; MTX = methotrexate; MMF = mycophenolate mofetil; ATG = anti-thymocyte globulin; PTCY = post-transplantation cyclophosphamide; GIAC = Granulocyte Intensified immunosuppression Antithymocyte globulin Combination pf peripheral blood and bone marrow; NR = nor reported; IS immunosuppression; pDLI = prophylactic donor lymphocyte infusion; CIR = cumulative incidence of relapse; LFS = leukaemia free survival; OS = overall survival; NRM = no-relapse mortality.

**Table 2 cells-13-00755-t002:** Retrospective studies with sequential protocols.

	N	Age	Inclusion Criteria	% Blasts (Median)	Donor	Sequential CT	Rest	Conditioning	GvHD Prophylaxis	IS Tapering	pDLI	CIR	OS	LFS	NRM
Ringden 2017 [44]	267	51	PIF and relapse	NR	MRD/MUD	FLAMSA	3d	TBI4Gy, Cy BU-based PAM	CyA + MTX/MMF ATG	NR	no	48%@3y	30%@3y	26%@3y	26%@3y
Dulery 2018 [45]	72	54	PIF, first/second relapse	NR	MRD/MUD haplo	TEC	3d	BU6.4FLU	CyA + MMF ATG	+60	yes	38%@2y	57%@2y	NR	24%@2y
Steckel 2018 [46]	292	56	Primary refractory Untreated relapse	32%	MRD/MUD	PAM140	5d	TBI8Gy/FLU TREO/FLU	CyA ± MTX/MMF ATG	NR	no	34%@1y	34%@3y	31%@3y	36%@1y
Saraceni 2019 [47]	856	51–58	PIF and relapse	NR	MRD/MUD	FLAMSA / /	NR	BU/TBI-based TREO/FLU TBF	CyA + MTX/MMF ATG	NR	yes	53%@2y 46%@2y 54%@2y	34%@2y 37%@2y 24%@2y	27%@2y 22%@2y 29%@2y	20%@2y 26%@2y 24%@2y
Rodrìguez-Arbolì, 2020 [47]	1018	39	PIF and relapse	NR	MRD/MUD	FLAMSA	NR	TBI-based CT-based MAC	CyA+ MTX/MMF ATG	NR	NR	55%@2y 53%@2y 51%@2y	36%@2y 50%@2y 33%@2y	27%@2y 40%@2y 30%@2y	18%@2y 7%@2y 19%@2y
Le Bourgeois 2020 [48]	131	52	PIF and relapse	NR	MRD	ClofaARAC	3d	BU9.6CY	CyA + MTX/MMF ATG	NR	no	45%@2y	38%@2y	29%@2y	35%@2y
Sockel 2022 [49]	173	56	Relapse (36%) first line	10%	MRD/MUD haplo	ClofaARAC	/	FLU-PAM Clofa-PAM	CyA + MTX/MMF PTCY	NR	no	30%@4y	43%@4y	NR	36%@4y
Guijarro 2022 [50]	140	55	PIF or relapse	20%	MRD/MUD haplo	FLAG-IDA	3d	PAM140 mg/m^2^	CyA + MTX/MMF ATG PTCY	NR	no	30%@5y	25%@5y	NR	45%@5y
Weller 2022 [51]	114	60	PIF or relapse	17%	MRD/MUD haplo	FLAMSA	3d	RIC	CyA + MTX/MMF ATG/PTCY	+90	yes	41%@2y	45%@2y	46%@2y (no DLI) 70%@2y (DLI)	27%@2y

TEC = thiotepa 5 mg/kg, etoposide 400 mg/m^2^, cyclophosphamide 2 g/m^2^. PAM = melphalan; PIF = primary induction failure; MRD = matched related donor; MUD = matched unrelated donor; mMUD = mis-matched unrelated donor; MAC = myeloablative conditioning regimens; RIC = reduced intensity conditioning regimens; FK = tacrolimus; CyA = cyclosporine A; MTX = methotrexate; MMF = mycophenolate mofetil; ATG = anti-thymocyte globulin; PTCY = post-transplantation cyclophosphamide; NR = nor reported; IS immunosuppression; pDLI = prophylactic donor lymphocyte infusion; CIR = cumulative incidence of relapse; LFS = leukaemia free survival; OS = overall survival; NRM = no-relapse mortality.

**Table 3 cells-13-00755-t003:** Prognostic factors retrieved from standard conditioning studies.

	OS	HR	LFS	HR	CIR	HR	NRM	HR
Duval 2010 [42]	Duration of first CR < 6 months	1.26	Not done		Not done		Not done	
	Duration of first CR > 6 months	0.83						
	Blasts PB at transplantation	1.48						
	HLA familiar other related	1.48						
	mMUD	2.21						
	KPS > 90%	0.65						
Craddock 2011 [36]	>3 CT	1.66	>3 CT	1.63	Not done		Not done	
	BM blast > median (38%)	1.49	BM blast > median (38%)	1.53				
	Recipient CMV+	1.63	Recipient CMV+	1.67				
Hemmati 2015 [37]	Not done		BM blast > 20%	1.58	BM blast > 20%	1.7	Not done	
			Any cGVHD	0.21	Any cGVHD	0.18		
					Any aGVHD	0.39		
Nagler 2015 [43]	Second relapse	1.5	Second relapse	1.54	First relapse	1.24	CYTBI	0.69
					Second relapse	1.73	Recipient age	1.24
Todisco 2017 [33]	>2 CT	1.87	Not done		Not done		Not done	
	BM blasts ≥ 25%/any level in PB	1.75						
	KPS < 90%	1.43						
	Recipient age > 60	1.77						
	Int/adverse cytogenetic	1.44						
Nagler 2022 [39]	Recent period HSCT	0.86	Recent period HSCT	0.87	Recent period HSCT	0.85	mMUD 9/10 Recipient CMV+	1.31 1.39
	Recipient age	1.05	mMUD 9/10	1.14	Recipient age	0.94		
	mMUD 9/10	1.2	Recipient CMV+	1.13	TBI	1.2		
	Recipient CMV+	1.2	Relapse	1.1	Relapse	1.21		
	Poor Cytogenetic	1.33	Poor Cytogenetic	1.51	Poor Cytogenetic	1.96		
Yanada 2022 [38]	Recipient age 40–49 y	1.29	Not done		PS 2–4	1.13	Recipient age 40–49 y	1.38
	Recipient age 50–59 y	1.52			Poor cytogenetic	1.6	Recipient age 50–59 y	1.72
	Recipient age > 60 y	1.74			Unevaluable cytogenetic	1.23	Recipient age > 60 y	2.25
	Recipient male	1.22			PB blasts 1–4%	1.29	Recipient male	1.23
	PS 2–4	1.79			PB blasts 5–19%	1.49	PS 2–4	1.21
	PIF	0.93			PB blasts > 20%	1.77	Poor cytogenetic	0.84
	Poor cytogenetic	1.66			CB	0.8	PB blasts > 20%	0.85
	Unevaluable cytogenetic	1.5			FK-based prophylaxis	0.91	CB	1.20
	PB blasts 1–4%	1.18			Year transplant 2016–2020	0.821	Year transplant 2016–2020	0.781
	PB blasts 5–19%	1.38						
	PB blasts > 20%	1.75						
	Year transplant 2011–2015	0.83						
	Year transplant 2016–2020	0.74						

PIF: primary induction failure; UR: untreated relapse; SR: secondary relapse; MAC: myeloablative conditioning; RIC: reduced intensity conditioning. MUD, matched unrelated donor; MRD, matched related donor; PS, performance status; KPS: Karnofsky performance status; PB: peripheral blood; BM: bone marrow; mMUD = mis-matched unrelated donor; CB: cord blood.

**Table 4 cells-13-00755-t004:** Prognostic scores from conventional allo-SCT studies.

	N	Score Components and Points	Score	OS	LFS
Duval 2010 [42]	1673	PIF or CR duration > 6 M = 0 CR duration < 6 M = 1 Cytogenetic good/INT = 0 Cytogenetic poor = 1 HLA match = 0 mMUD = 1 Haplo = 2 Circulating blasts yes = 1 KPS < 90 = 1	0 (1 point) 1 (1 point) 2 (2 points) ≥3 (3 points)	42% 28% 15% 6%	Not done
Craddock 2011 [36]	168	>2 CT = 1 BM blasts > median = 1 R CMV+ = 1	0 (0 point) 1 (1 point) 2 (2 points) 3 (3 points)	44% 24% 10% 0%	40% 24% 12% 0%
Todisco 2017 [33]	227	>2 CT = 1 BM blasts >25%/circulating any level = 1 Age > 60 = 1 Cytogenetic poor = 1	0 (0–1 points) 1 (2 points) 2 (>2 points)	32%@3y 10%@3y 0%@3y	Not done

PIF: primary induction failure; CR: complete remission; mMUD: mis-matched unrelated donor; KPS: Karnofsky performance status; CT: chemotherapy; R: recipient.

**Table 5 cells-13-00755-t005:** Significant prognostic factors retrieved in sequential conditioning studies.

	OS	HR	LFS	HR	CIR	HR	NRM	HR
Ringden [48]	In vivo T-cell depletion	0.46	In vivo T-cell depletion	0.49	UD (vs. MRD)	0.6	Recipient age UD (vs. MRD) In vivo T-cell depletion	1.33 1.96 0.35
Steckel [44]	Recipient age > 59 y Time from diagnosis to HSCT < 9M >20% blasts (BM/PB) mMUD	adverse adverse adverse adverse	Not done		Not done		Recipient age > 59 y HCT-CI ≥ 2 mMUD Infection before conditioning	NR
Dulery [54]	None		Not done		Not done		KPS < 90%	3
Saraceni [47]	KPS ≥ 80% Recipient CMV+	0.7 1.3	Recipient CMV+	1.4	Age Relapse (vs. primary refr) Recipient CMV+	0.9 1.3 1.3	Recipient age mMUD	1.3 1.8
Rodrìguez-Arbolì [45]	FLAMSA-CT Adverse/failed cytogenetic Second relapse KPS > 90% UD (vs. MRD)	0.65 2.13/2.02 1.88 0.54 1.23	FLAMSA-CT Adverse/failed cytogenetic Second relapse KPS > 90%	0.73 2.07/1.99 1.78 0.64	Recipient age Adverse/failed cytogenetic Second relapse KPS > 90%	0.85 2.93/2.55 1.94 0.67	FLAMSA-CT Recipient age KPS > 90% UD (vs. MRD)	0.4 1.28 0.57 1.94
Le Bourgeois [50]	CMV−/−	1.75	CMV−/−	1.71	CMV−/−	2.49	None	
Sockel [9]	>20% blasts (BM) Recipient age AML therapy related	1.8 1.26 2.10	Not done		5–20% blasts (BM) >20% blasts (BM)	1.18 1.24	AML therapy related Recipient age UD	3.39 1.26 2.46
Guijarro [49]	Recipient age > 55 y	2.56	Not done		Adverse cytogenetic	2.65	Recipient age > 55 y	2.4

UD: unrelated donor; mMUD: mis-matched unrelated donor; MRD: matched related donor; KPS: Karnofsky performance status; OS: overall survival; LFS: leukaemia free survival; CIR: cumulative incidence of relapse; NRM: no relapse mortality; HR: hazard ratio.

**Table 6 cells-13-00755-t006:** Main results from prospective studies.

	N	Median Age	Inclusion Criteria	% Blasts at ALLO	CR	Sequential	Rest	CTX	Donor	GVHD Prophylaxis	IS Tapering	Prophylactic DLI	CIR	OS	LFS	NRM
Schmid 2005 [52]	75	52 y (18–65)	No response to HD ARAC Relapse 3 M after CR Second relapse Delayed response to IC Relapse after auto Secondary AML/MDS	NR	11%	FLAMSA-RIC *	3 days	TBI4Gy, Cy	MRD 49% MUD 51%	CSA day −1 MMF Day 0 ATG	CSA day +60 to 90 MMF day +50	24% Day +120 Median day +160	20%	42%@2y	40%@2y	33%@1y
Schmid 2006 [57]	103	51 y (18–68)	PIF after ≥2 IC Relapse 6 M after CR Refractory to salvage IC ≥2nd relapse	30% (0–90%)	4%	FLAMSA-RIC *	3 days	TBI4Gy, Cy	MRD 40% MUD 60%	CSA day −1 MMF Day 0 ATG	CSA day +60 to 90 MMF day +50	24% Day +120 Median day +159	37%	40@2y	39%@2y	17%@1y
Middeke 2016 [58]	84	61 y (40–75)	PIF after ≥2 IC Relapsed	54% (5–92%)	None	ClofaARAC	ALLO in aplasia	ClofaPAM	MRD 18% MUD 54% mMUD 29%	CSA day −1 MMF Day 0 ATG (only in mMUD)	Not reported	No	26%@2y	43@2y	DFS 52%	23%@2y
Jaiswal 2016 [59] ^	41	26 y (2–65)	PIF after ≥2 IC Relapsed refractory	14–16% (5–65%)	None	no	/	BUFLUPAM ^	Haplo	PTCY day +3 and 4 CSA day +5 MMF day +5	CSA day +60 MMF from +14 to +21	90% Day +21, +35, +60	43% 21% with DLI	53%@18 M 70% with DLI 35% w/out	44%@18 M 62% with DLI 25% w/out	19%@1y
Mohty 2017 [60]	24	47 y (20–57)	PIF after 2 IC Persisting hypoplasia	20% (6–82%)	None	ClofaARAC-RIC §	3 days	BUCY	MRD 63% UD 37%	CSA day −1 MMF Day 0 (only in UD) ATG	CSA day +90 MMF +62 to 90	25% Day +120	54%@2y	38%@2y	29%@2y	12%@2y
Davies 2018 [55]	47	53 y (23–68)	PIF after 1 IC Relapse 6 M after CR	NR	None	DaunoARAC-RIC	3 days	FLUCY	MRD 49% MUD 51%	CSA day −1 Short MTX	CSA day +90	No	30%@3y	39%@2y	39%@2y	35%@1y
Bonifazi 2022 [61]	101	54 y (16–69)	PIF Relapse	30% in PB	None	no	/	TBF °	MUD 57% Haplo 38% CB 9%	ATG-based PTCY-based	NR	No	53%	19%@2y	19%@2y	35%@1y

IC: induction chemotherapy; PB: peripheral blood; CTX: conditioning regimen; IS: immunosuppression; OS: overall survival; LFS: leukaemia-free survival; CIR: cumulative incidence of relapse; NRM: no relapse mortality; HR: hazard ratio. * RIC consisted of total body irradiation (4 Gy), ATG, and cyclophosphamide. § The RIC consisted of busulfan 6.4 mg/kg + cyclophosphamide 60 mg/kg. ^ excluding 10 patients treated with a nonmyeloablative conditioning regimen. The MAC consisted of fludarabine 150 mg/m^2^, busulfan 9.6 mg/kg, and malphalan 140 mg/m^2^. ° MAC consisted of thiotepa 10 mg/kg, busulfan 9.6 mg/kg, and fludarabine 150 mg/m^2^.

**Table 7 cells-13-00755-t007:** Factors predictive of outcome in prospective studies.

	N	Sequential	pDLI	OS	LFS	NRM
Schmid 2005 [52]	75	FLAMSA-RIC	yes	CD34	CD34	CD34 MRD
Schmid 2006 [57]	103	FLAMSA-RIC	yes	CD34 <2 IC	CD34 <2 IC	/
Middeke 2016 [58]	84	ClofaARAC	No	Age No response on day +15	/	/
Jaiswal 2016 [59]	41	no	yes	MAC + pDLI	MAC + pDLI	/
Mohty 2017 [60]	24	ClofaARAC-RIC	yes	/	/	/
Davies 2018 [55]	47	DaunoARAC-RIC	No	cGVHD	cGVHD	/
Bonifazi 2022 [61]	101	no	No	HCT-CI > 0 Low risk cytogenetic	/	HCT-CI > 0

pDLI = prophylactic DLI; HCT-CI: hematopoietic cell transplantation-comorbidity index; MAC: myeloablative conditioning regimen; cGVHD: chronic graft versus host disease.

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
