# Peer review of "Allogeneic Stem Cell Transplantation in Refractory Acute Myeloid Leukaemia"

_cells, 2024, doi:10.3390/cells13090755_

Round 1
Reviewer 1 Report
Comments and Suggestions for Authors
As authors state, the present review describe and comment the results from retrospective and prospective studies about allogeneic transplant in refractory AML patients published during the 2010-2023 period. Based on those findings they speculate how to modify different parts of transplant procedure to maximize its therapeutic effect.
I have some comments and suggestions for authors
The way that authors describe the allogeneic transplant results in refractory AML patients is not easy to follow. This is a common problem in many similar manuscripts. It is a shame that authors have not been more ambitious and they would have done a meta-analysis with the reviewed studies. It would allow more valid conclusions to be drawn for authors and readers.
1. In the abstract section, I suggest to authors first describing succinctly the more recent findings in the field and then describe the purpose of review.
2. Authors state in the results section from retrospective studies that they consider only published studies that involve more than 100 patients. I think it is a description of criteria used to for reviewing the papers. I think that authors should describe the method used for searching ( key words, platforms, etc) and how the studies were selected for reviewing. In other words, a method and study design section describing this procedure.
3. Other important problem with this review is the definition of refractory AML. I realize that definition of refractory AML has evolving over time. However, authors should clearly state which has been the definition used when they selected the papers for reviewing.
4. The last part of manuscript should address some significant remaining challenges such as how to include development of synergistic combination therapies ( targeted therapies, immunologic etc) and redefining the role and timing of allogeneic transplant in refractory AML patients.
Author Response
Dear Reviewer
thank you for the time spent to reise the manuscript and for constructive observations.

Reviewer 2 Report
Comments and Suggestions for Authors
In this review paper, the authors are providing comprehensive information about how to tackle refractory acute myeloid leukemia (AML), summarizing previously published data from retrospective and prospective studies. Indeed, as the authors mention, substantial number of patients whose AML is recurrent and refractory to conventional chemotherapies represent unmet medical needs in the current clinical hematology. From this point of view, the aim and attempt of this paper is valuable.
Comments
1. Some parts of the manuscript seem complicated and hard for readers to go through. This is partly due to abbreviations without explanations or full spelling (e.g FLAMSA, mMUD…).
2. Figure 1 (written as Figure S1 in the text) needs a proper legend that explains its contents in detail (e.g criteria for low-intermediate and high-risk of ELN classification) and delivers its main claim.
Comments on the Quality of English LanguageThe editing by native English speaker would ameliorate readability.
Author Response

(The authors gave the same response as above.)

Reviewer 3 Report
Comments and Suggestions for Authors
See pdf with all the comments

Some of the issues with English language have been corrected, but still a proof-reading is needed.
Author Response

(The authors gave the same response as above.)

Round 2
Reviewer 1 Report
Comments and Suggestions for Authors
Dear Authors
After reading the new manuscript I think that the queries have been resolved favorably
Author Response
Thank you for your revision and to consider the manuscript improved.
Reviewer 3 Report
Comments and Suggestions for Authors
- the answers to the remarks and questions are generally quite brief and could have been more thorough
Comments on the Quality of English Language- still some mistakes in English language, sentences that do not fit
- answer to the last question on confounding factors between MAC and FLAMSA-RIC, has a lot of strange sentences, that are hard to understand, and should be adapted, also in the article.
Author Response
Rewiever comment: Answer to the last question on confounding factors between MAC and FLAMSA-RIC, has a lot of strange sentences, that are hard to understand, and should be adapted, also in the article
Thank you again for the time spent to revise the manuscript.
Regarding the question above, in the text and in the answer to the specific question, we underlined that the paper analyzed was a retrospective one with the usual limits linked to selection bias. Of course the Authors tried to control for the differences between the cohorts using the MV analysis. The results reported were only suggestive.
Regarding the last paragraph on the predicitve factors of outcomes reported in the Table 7, we modified a little bit to improve the readability.